# S2-Attention: Hardware-Aware Context Sharding Among Attention Heads

## Abstract

Sparse attention, which selectively attends to a subset of tokens in the context, has been an established approach to enhance the efficiency of Transformers. However, its theoretical reduction in FLOPs has rarely translated into wall-clock speed-up over its dense attention counterparts, mainly due to the lack of hardware-level optimizations like FlashAttention. Meanwhile, it remains unclear whether sparse attention can maintain the model's quality at a scale of today's large language models (LLMs), and how this can be achieved. This paper presents Sparsely-Sharded(S2) Attention, a Triton library that provides kernel optimization for sparse attention customizable at both per-head and per-context-range levels. S2-Attention enables the exploration of novel and high-performance sparse attention techniques, which we demonstrate through extensive ablations across a wide range of sparse attention designs at various model scales. From these insights, we present several basic guidelines to design sparse attention that can achieve not only practical efficiency improvements, but also strong performance on downstream tasks. To achieve high parallelization and optimized memory IO, sparse attention should **shard the context heterogeneously across attention heads**, where each head attends to a different subset of tokens while **collectively covering the full context**. Meanwhile, we find hybrid architectures combining sparse and dense attention particularly beneficial in practice. These design choices lead to a novel sparse attention architecture, which we evaluate with 1.3B, 7B models. It achieves wall-clock speedup of 8.79X, 15.87X, 25.3X compared to the strong FlashAttention-2 baseline with strong downstream performance on-par with full attention and perfect retrieval performance at a 128k context length. In inference, for 7B models, our model, with the help of our S2-Attention kernel, achieves 4.5x speed-up compared to dense counterparts. S2-Attention will be released with easy-to-customize APIs for direct usage in Megatron and vLLM. We hopet they can help broader grounding and exploration of sparse training and inference for the community.

## 1 Introduction

Transformer-based LLMs have opened up fresh opportunities to both research and applications (OpenAI, 2023; Touvron et al., 2023). Their quadratic overhead imposes prohibitive overhead in training and serving these models. For example, training Llama 2 (Touvron et al., 2023) 70B with a 4K context length on 2 trillion tokens takes 23 days on 2048 A100 GPUs Rucinski (2024). When serving the model, the model's KV cache consumes 343 GB GPU memory with a batch size 32 and 4K context length. There is an urgent need to train LLMs efficiently and serve them cost-effectively.

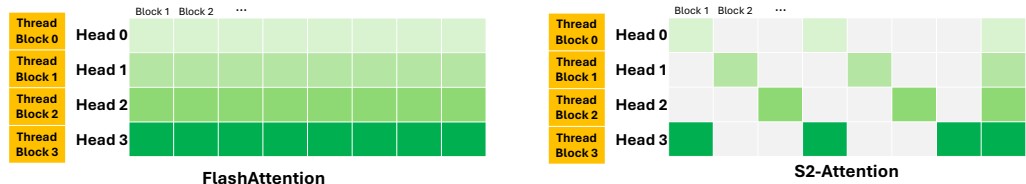

Figure 1: Illustration of S2-Attention with four attention heads on a hypothetical GPU with 4 thread blocks. Each attention head is allocated with a shard of the context.

Many works aim to improve efficiency of attention through various **sparse attention** techniques (Tay et al., 2023; Child et al., 2019; Beltagy et al., 2020; Zaheer et al., 2020), where only a subset of the tokens in the context are attended to. Despite their promising on-paper FLOP savings compared

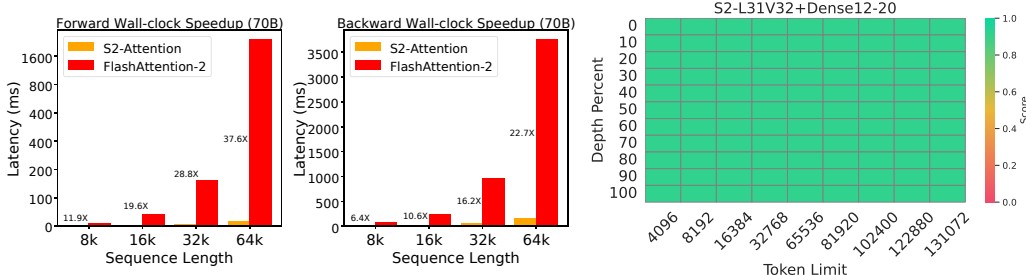

(a) Attention Latency Benchmark over FlashAttention-2.

(b) Perfect 128k Needle in a haystack.

Figure 2: Training Efficiency and long-context analysis of S2-Attention. Our model, implemented with our kernel, achieves substantial reduction in latency compared to FlashAttention-2 (a). It also achieves perfect retrieval performance at a 32K context length (b).

to full-context **dense attention**, these methods often fail to deliver real-world efficiency gains. As pointed out by the seminal work FlashAttention (Dao et al., 2022; Dao, 2023), GPU memory access, rather than computation, is the primary bottleneck for attention. Dense attention has benefited from CUDA-level implementations specifically optimized for more efficient memory IO, an significant optimization that sparse attention methods have yet to receive. The absence of a flexible, efficient, and easy-to-use library for optimized implementations for sparse attention has become a major roadblock for research in this area, slowing the progress in improving LLMs' training and serving efficiency.

With Sparsely-Sharded(S2) Attention, we aim to bridge this gap. S2-Attention is a Triton library that provides kernel optimization for sparse attention. It is highly flexible, allowing practitioners to explore various sparse attention strategies and customize different attention patterns across attention heads and context ranges. The main challenge in implementing a fused kernel for general sparse attention arises from the sparsity itself. Tiling the Q, K, and V matrices in sparse attention results in idle threads within each tile, as parts of the context are not attended to, leading to reduced thread and SRAM usage while decreasing the FLOP reduction benefits. S2-Attention addresses this by efficiently tracking the KV usage patterns and dynamically merging sparse query blocks that share common KVs into the same tile, ensuring each tile is fully utilized regardless of sparsity granularity. This approach substantially improves SRAM utilization and minimizes redundant KV loading. S2-Attention is applicable in both training and inference, substantially lowering the barrier to exploring novel sparse attention architectures, which we explore in Sections 4 and 5.

Based on the insights from our systematic exploration of sparse attention strategies, we propose a novel architecture based on three key principles: (1) Efficient sparse attention must be designed with hardware and software systems in mind. Many sparse variants are incompatible with accelerator parallelization and efficient memory access patterns, making it challenging to translate their FLOP savings into real-world efficiency gains. (2) Our approach uses a hardware-friendly sparse attention pattern that shards the context heterogeneously among attention heads. Each head applies a unique stride attention pattern to cover a different subset of tokens, while, collectively, they cover the full context (Figure1). (3) To achieve strong performance on challenging long-context tasks, the model must have direct access to all tokens, at least at certain layers. A hybrid design that combines sparse and dense attention balances efficiency and performance.

To evaluate S2-Attention and our proposed hybrid architecture, we pre-train a suite of models at 1.3B, 7B scales with different sparse attention, and compare their quality to the dense attention baseline. Our results show that the 7B hybrid architecture matches the performance of dense attention while achieving a 2.5X training speed-up and 4.5X inference speed-up. Moreover, we extend the 1.3B models to a 32k context length, and 7B models to 128k. We show that the hybrid structures can achieve perfect Needle in a Haystack retrieval performance. Compared to FlashAttention-2 (Dao, 2023), S2-Attention can achieve as much as 8.79X, 15.87X attention speed-up for 1.3B, 7B scale respectively, and 2.83X,2.47 e2e training time reduction.

S2-Attention is compatible with commonly used LLM frameworks including PyTorch, Megatron, HuggingFace, and vLLM. With its user-friendly APIs, importing and customizing S2-Attention take no more than several lines of code. S2-Attention will be publicly released upon publication.

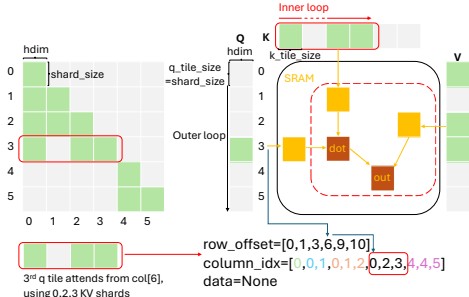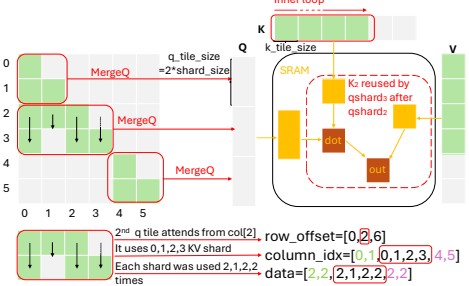

Figure 3: Illustration of S2-Attention System Implementation. Left: Naive CSR-based strategy can only efficiently handle very coarse sparse pattern that keep or mask context chunks of block size. Right: MergeQ adaptively merge smaller query shards to tile size, significantly improving the system efficiency and flexibility.

## 2 RELATED WORKS

**Efficient Transformers** Numerous of attempts have been made to make the training and serving of LLMs more efficient. FlashAttention family (Dao et al., 2022; Dao, 2023) are the most widely-adopted attention acceleration framework. Dao et al. (2022) breaks down the attention computation into smaller block-wise computation to reduce the IO between SRAM and HBM. Dao (2023) further improve the performance by reducing the non-matmul operations, extra parallelization over sequence length, and better warp organization. Our implementation is based on the FlashAttention kernel and we leverage its parallelization over the number of heads.

**Sparse Self-Attention** reduce the computational complexity of self-attention by only attending to a subset of tokens(Child et al., 2019; Katharopoulos et al., 2020; Kitaev et al., 2020; Zaheer et al., 2020; Beltagy et al., 2020). Child et al. (2019) factorized attention computation into local and stride patterns to reduce the computation. Ho et al. (2019) saves computation by applying multiple attentions along each axis of the input tensor. Beltagy et al. (2020) adds more sparsity on top of Child et al. (2019) by taking advantage of the receptive field accumulation with dilated sliding window. Zaheer et al. (2020) replaces the one-to-all attention in transformers with sliding window attention and random attention. Qiu et al. (2020) chunks the sequence into blocks, which reduces FLOPs by performing attention on a larger granularity. However, many of these methods can't bring wall-clock speed-up due to ignorance of realistic memory access cost (Dao et al., 2022). Meanwhile, we argue that some sparsity patterns are not inference-friendly for generative models due to inefficient KV caching. Also, as they underutilized the parallelization over heads, there's actually room to further reduce FLOPs while improving context coverage. We will discuss this further in the following sections.

## 3 S2-ATTENTION: EFFICIENCY AND CUSTOMIZATION

In this section, we present the system implementation of S2-Attention. We first briefly cover the basic GPU memory and execution hierarchy. Based on the discussion, we show why sparse attention will have suboptimal performance when naively following the same tiling strategy of FlashAttention series. We then propose Merge-Q in Section 3.3, a technique that can significantly improve the kernel efficiency while allowing more fine-grained sparsity customization by adaptive merging query segments that shares KV.

### 3.1 PRELIMINARIES

GPU threads can access several types of memory, including registers, on-chip shared-memory (SRAM), and the global high bandwidth memory (HBM). Access times vary greatly, with registers being the fastest and the HBM being the slowest. Inefficient memory access patterns, such as frequent I/O to HBM, can significantly hurt efficiency. CUDA organizes threads into blocks, and blocks are further divided into warps, a group of 32 threads. Threads within a block share the data through SRAM. It is desirable that different threads in the same warp take the same execution path since otherwise efficiency will be hurt due to warp divergence. Besides, thread block size should be

sufficiently large to achieve good utilization and load balancing. Lastly, a tile is a portion of the data assigned to a thread block to handle. For clarity, we take tile size as block size. FlashAttention imrpvoes efficiency by minimizing HBM I/O, tiling the Q, K, V matrices into chunks that fit into SRAM for efficient computation (Dao et al., 2022), a principle we follow below.

**Why does sparse attention not fully benefit from fused kernels?** When tiling the Q, K, and V matrices, some parts of the context are not attended to by the sparse attention. As a result, some warps may be assigned with tiles where only a small subset of threads are active, reducing the utilization and overall throughput. The issue is easier to handle if the sparsity is enforced at a granularity close to thread block size, e.g., keep or mask for every 64 tokens, as the non-sparse portion can occupy the whole block without warp divergence. However, for finer granularities, this strategy will not work, which S2-Attention aims to address.

## 3.2 Overall Kernel Design

As shown in Figure3, for each query vector $q$, we iterate through the $K$ tiles in SRAM to compute $\text{softmax}(qK^\top)$. Unlike dense attention, which uses the entire $K$ matrix, we only consider the keys specified by the sparse attention pattern, defined by an attention mask. This mask is stored in a Compressed Sparse Row (CSR) format to optimize memory efficiency.

Given a sequence of $N$ tokens, without loss of generality, we first chunk context into a list of blocks of size $S \in [1, N]$, which gives us $B = ceil(N/S)$ blocks. The attention mask, M, is then a B*B matrix. Starting from here, to avoid ambiguity with thread blocks in kernel, we refer context blocks as **shards**. We use $qshard_i$ and $kshard_i$ to denote the ith query shard and ith key shard.

For simplicity, we first consider a scenario where the shard size is the same as the tile size of Q, K, V. As shown in the left side of Figure 3, given a attention mask M, we first transfer it into the CSR format. Each row can be viewed as the attended KV positions for each qshard, where the corresponding index is specified in the column_idx array. row_offset serves as pointers for each key to find its attended KV in column_idx. Then when computing $qshard_i K^T$ in SRAM, we only need to loop the necessary ones according to column_idx.

## 3.3 Merge-Q

The implementation above works and can already generate expected speed-up. However, the premise here is that the context shard size is the same as the tile size to avoid divergence. Ideally, we want the context shard size to be smaller to customize more fine-grained sparsity, but kernel block size are usually required to be much larger to increase GPU efficiency. Such conflicts would greatly limit the granularity of sparsity patterns S2-Attention can support.

Therefore, it is critical to decouple the context shard size and the kernel tile size in the implementation. To achieve that, we design Merge-Q operation, which merges the context shards with the same K/V blocks but different q shards into a mega shard that forms a tile in the kernel. Figure 3(b) illustrate a simpler case, where we only merge neighboring q shards. Thanks the insight in section 4.1 on KV-efficient sparsity, we only need to know the number of non-empty shards within the mega shard regardless of where the empty shards are, as empty blocks are either at the end or out of the causal region. For implementation, we twisted the CSR definition to use the data array to store how many times a $kshard[i]$ is used in the mega shard. This help improve the reuse of those K/V shards, as now instead of reloading them into chip multiple times for different qshard, the kernel now only needs to do it once.

Merge-Q allows us to support shard size as small as 16. As shown in Figure 4, we see significant improvement after merging multiple small blocks in the Q dimension. For shard size larger than the best kernel block size, we simply break the larger shard to a few of shards that are optimal for GPUs.

## 3.4 Split-D

FlashAttention load the whole head dimension(D) at once to SRAM, which is intuitive as the whole vector is needed to compute the attention scores. However, in our experiment, we found that split over the D dimension in many cases is beneficial. We suspect that this is because Split-D reduce SRAM usage and enable a larger degree of software pipelining. Interestingly, we also find that with

shard size of 64, Split-D only helps when head dimension is 128, the most commonly used case, while has no additional benefit when head dimension is 64 or 256. We suspect that this is due to under-tuned hyper-parameters like kernel tile sizes.

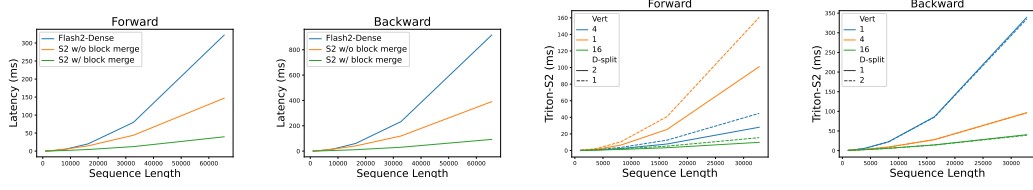

(a) Forward w/ Merge-Q.    (b) Backward w/ Merge-Q.    (c) Forward w/ Split-D.    (d) Backward w/ Split-D.

Figure 4: Benchmark Latency Improvement with Merge-Q and Split-D.
All experiments are done using (batch, heads, head dim) = (4, 16, 128) on an A100 80GB-SXM GPU.
In (a) and (b), (shard size, vertical stride) = (16, 16). In (c) and (d), the shard size is 64.

With the optimized kernel library, the community can customized fine-grained sparsity patterns while enjoying the speed-up. However, what sparsity pattern can bring speed-up without quality degradation remains unknown. We seek to answer this question and derisk architecture choices with out insights and experiments.

## 4   S2-ATTENTION: INSIGHTS, FORMULATION, AND SPARSITY COOKBOOK

In this section, we first discuss which kind of sparse attention patterns are efficient for decoder-only LLMs. Building on these insights, we discuss strategies to improve context coverage while achieving more savings in memory and FLOPs. Lastly, we discuss general guidelines to design efficient and effective sparse attention.

### 4.1   KV-EFFICIENT SPARSITY

KV caching is a primary memory bottlenecks for decoder-only LMs at inference time. Many existing sparse attentions determine which tokens to attend to based on relative distances (Child et al., 2019; Zaheer et al., 2020; Beltagy et al., 2020). However, these approaches are not GPU memory-efficient during decoding, making it difficult to translate their FLOP savings into real-world efficiency gains. Figure 5(a) provides an illustrative example. The main issue is that, for such sparse attention, KV not used in earlier decoding steps might be required in later ones, making memory management more challenging. Despite the nearly 50% memory saving on paper, it actually requires storing the full KV cache in practice, resulting in zero memory savings.

In contrast, Figure 5(b) illustrates a sparse attention that can achieve memory saving in practice. The key is that the stored KV cache is reused across several decoding steps but is no longer needed in future steps, and thus can be evicted, freeing up the GPU memory.

The comparison between these two approaches leads to the following general principle of designing KV-efficient sparse attention. For $\forall j \geq i$, $l \geq 1$,

$$(\boldsymbol{k}_i, \boldsymbol{v}_i) \text{ is attended by } \boldsymbol{q}_{j+l} \implies (\boldsymbol{k}_i, \boldsymbol{v}_i) \text{ must also be attended by } \boldsymbol{q}_j. \tag{1}$$

Otherwise, $\boldsymbol{k}_i$ and $\boldsymbol{v}_i$ need to be stored at step $j$ for future generations, even it is not used at step $j$. This basically means that once a token is pushed into KV-cache, it will remain being used until evicted. Intuitively, in the attention pattern matrix, we shall see continuous "vertical lines" as shown in Figure 5(b). This means the sparse patterns should be based on absolute positions rather than relative ones, except for consecutive local context (e.g., left figure in Figure 5(b)).

It is important to point out that all eviction strategies targeting inference ((Zhang et al., 2023; Liu et al., 2023; Ge et al., 2024)) are by nature KV-cache efficient, as they satisfy definition (1). However, the focus of KV-cache pattern is mostly on training, to avoid patterns that do not lead to KV-cache saving like the one in Figure 5(a).

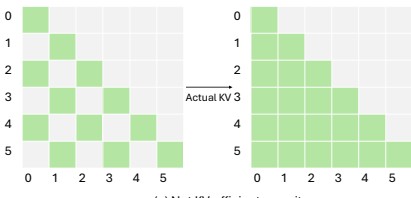
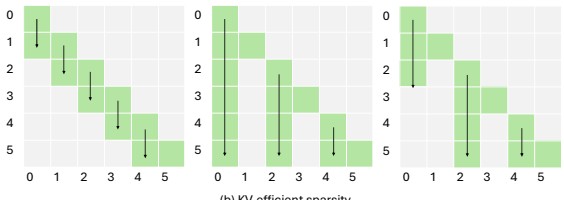

(a) Not KV-efficient sparsity          (b) KV-efficient sparsity

Figure 5: **(a):** The dilated attention based on relative position as an example of sparse attention that is not KV-efficient. E.g., step 5 attends to KV at positions 1, 3, 5, while step 4 attends to 0, 2, 4. This results in requiring full KV cache. Although it suggests nearly 50% memory savings on paper, it actually requires storing the full KV cache in practice. **(b)** All these attention patterns are KV-efficient, as they get pushed to KV-cache when first encountered at decoding, then continuously being attended for several steps before it finally gets evicted (e.g., all tokens in left figure, and token 0 in right figure) and never gets attended again, or remained attended for all future tokens (e.g., tokens 0, 2, 4 in middle figure and tokens 2, 4 in right figure). The arrows show that they all share a "vertical line" pattern.

## 4.2 HETEROGENEOUS CONTEXT SHARDING

A simple pattern that meets the principle above is a combination of fixed positions, dubbed **vertical stride**, and a local window. Each attention head accesses to a partial context at any given step. We have different heads should attend to different subsets of equal-sized positions. They also, collectively, cover the full context. We dub this method **heterogeneous context sharding**. Compared to using the same pattern across all heads, our heterogeneous sharding comes with the same overhead; but it ensures that the model always has access to the full context. The flexibility of our S2-Attention kernels facilitate us to implement this strategy efficiently. As we will show in experiments, this ensures that the model can achieve strong long-context performance while maximizing the efficiency gains. Figure 1 provides an illustrative example.

For a transformer with $H$ heads, we chunk a sequence of $N$ tokens into $B = ceil(N/S)$ shards. Among the total $B$ blocks, we take the most recent $N_l$ blocks as local blocks and set the rest as remote blocks. We denote the remote stride as $v_r$. In practice, $v_r$ is set to be a factor number of attention heads to satisfy the "union as full context" constraint. For each head $h \in \{1, ..., H\}$, we set a unique position offset $o_h$. For simplicity, we set $o_h = h$ here, which means the offset of each head equals its head index.

Then for each head $h$, its block attention mask $M^h$ in $B \times B$ dimension is as follows:

$$M_{i,j}^h = \begin{cases} 1 & q_i - k_j < N_l, \\ 1 & (k_j - o_h) \in v_r \mathbb{Z}_{\geq 0} \ \& \ q_i - k_j \in [N_l, B) \\ 0 & \text{otherwise,} \end{cases} \quad (2)$$

where $x \in m\mathbb{Z}_{\geq 0}$ mean $x$ is 0 or a positive multiple of $m$.

The first constraint determines the shards within the local window. The second constraint controls what shard each head attends to outside the local window. Note that if the distance is beyond $B$, the attention is dropped, thus extends similar to a sliding window.

This local-stride pattern can be easily extended to have multiple strides in different context ranges. With blocks $N_l < N_{r_1} < N_{r_2} < B$, and $v_{r_1} < v_{r_2}$, we define block attention mask as

$$M_{i,j}^h = \begin{cases} 1 & q_i - k_j \in [0, N_l), \\ 1 & q_i - k_j \in [N_l, N_{r_1}) \ \& \ (k_j - o_h^1) \in v_{r_1} \mathbb{Z}_{\geq 0}, \\ 1 & q_i - k_j \in [N_{r_2}, B) \ \& \ (k_j - o_h^2) \in v_{r_2} \mathbb{Z}_{\geq 0}, \\ 0 & \text{otherwise.} \end{cases} \quad (3)$$

To ensure that $M_{i,j}^h$ is KV-cache efficient, we requires that $v_{r_2} \in v_{r_1} \mathbb{Z}_{\geq 0}$ and $(o_h^2 - o_h^1) \in v_{r_1} \mathbb{Z}_{\geq 0}$.

### 4.3 Union Completeness, Hybrid Architecture

Despite **Heterogeneous Context Sharding**, we introduce two other insights that we empirically find useful for better model quality. With the formulation in Section 4.2, we can derive that the full context can be preserved if the vertical stride size is not larger than the number of attention heads. We denote such property as **Union Completeness**, when the union of all the attention heads' shards generate the full context.

As previous studies show (Huang et al., 2022; Lieber et al.), some attention layers are significantly denser compared to the others, with attention weights distributed near uniformly across all positions. Therefore, it is particularly beneficial to retain dense attention in these layers. This motivates us to explore a hybrid architecture that combines our efficient sparse attention in most layers with dense attention in others. We empirically find that our sparse attention strategy is highly effective, requiring only 1/6 of the attention layers to be dense to achieve strong retrieval performance with 128K-long contexts. More exploration is presented in our experiments.

## 5 Experiment

We evaluate the efficiency advantage and quality preserving S2-Attention can achieve. We first study the pre-training quality is Section5.1 and Section5.2. We then benchmark the kernel efficiency and end-to-end serving latency in Section5.3 and Section5.4. Lastly, we ablate a few sparsity heuristics for better understanding.

### 5.1 Benchmarking Model Pre-Training Quality

**Experiment Settings.** To study the S2-Attention, we first train a range of 1.3B Llama 2 structure models with 24 layers, 2048 hidden size with 16 heads, with max sequence length as 8192. We use the open-source FineWeb-Edu-350B Penedo et al. (2024) as the pre-training corpus. An OpenAI Tiktoken tokenizer with 100K vocabulary size is used to process the raw text. All model variations use batch size of 4M tokens for all sequence lengths and train for a total of 300 billion tokens. For hyperparameters, we use $\mu$P Yang et al. (2022) with a base shape of 256. A $\mu$P learning rate of 0.02 is used with linear decay and 0.5% of total training tokens for warmup. All models are evaluated after training on the total 300B tokens for one epoch.

**Downstream tasks.** We use a model with default attention as our baseline, denoted as Dense. To study our hybrid structure with heterogeneous sharding and union completeness, we control the FLOPs to be approximately equivalent. The total attended tokens is around 576 tokens, or 9 shards of 64 tokens. We use this to configure the sliding window attention (SWA), as the control set. We add different changes to SWA to see how they affect the training quality. The treatment sets are grouped into Homogeneoeous, Heterogeneous&Incomplete, and Heterogeneous&Complete for comparison.

From Table 1, we can observe the hybrid architectures shows promising results. As we can see from S2-L1V15+Dense in the last row, heterogeneous sharding with complete context and two dense layers give consistently best results across tasks, with minor gap from the default attention baseline while using only 18% FLOPs. Notably, in the Passkey Retrieval task, S2-Attention can achieve much better performance compared to the dense model. This observation work as an initial demonstration of the context understanding ability of the S2-Attention design. We'll discuss more the continual pre-training section.

Across Homogeneous and Heterogeneous settings, adding two dense layers will translate to significant performance bonus. Within the Homogeneous group, in addition to dense layers, we can observe adding attention sink can significantly boost training quality, compared to only using SWA. In the Heterogeneous&Incomplete group, the vertical stride size is bigger than the number of attention heads, making the context incomplete after union. For the Heterogeneous&Complete group, we tune the stride size and local window so that it's just covering the full context while having the same FLOPs as the SWA control set. When comparing the Incomplete group to the Complete group, we can see the benefits of making the union of context complete by limiting vertical stride size.

Table 1: Training quality evaluation. SWA refers to sliding window attention, while S2 refers to S2-Attention. L refers to number of local blocks. V refers to the vertical stride size. + sink refers attending to attention sink. + Dense refers to making the first two attention layers dense.

| Model | Passkey | WinoGrande | piqa | race | wikitext103(ppl) |
|---|---|---|---|---|---|
| Dense | 0.865 | 0.592 | 0.733 | 0.403 | 15.884 |
| **Homogeneous** | | | | | |
| S2-L9(SWA) | 0.334 | 0.547 | 0.705 | 0.363 | 21.997 |
| S2-L9 + Dense | 0.62 | 0.575 | 0.714 | 0.373 | 20.450 |
| S2-L9+sink | 0.560 | 0.566 | 0.721 | 0.380 | 21.037 |
| S2-L9+sink+ Dense | 0.771 | 0.577 | 0.728 | 0.388 | 18.503 |
| S2-L1V15 | 0.542 | 0.541 | 0.716 | 0.352 | 21.035 |
| S2-L1V15 + Dense | 0.741 | 0.568 | 0.713 | 0.349 | 20.579 |
| **Heterogeneous & Incomplete** | | | | | |
| S2-L2V18 | 0.630 | 0.565 | 0.728 | 0.357 | 20.502 |
| S2-L2V18 + Dense | 0.823 | 0.587 | 0.732 | 0.379 | 18.726 |
| S2-L4V25 | 0.612 | 0.542 | 0.720 | 0.352 | 20.875 |
| S2-L4V25 + Dense | 0.795 | 0.569 | 0.724 | 0.386 | 19.285 |
| **Heterogeneous & Complete** | | | | | |
| S2-L1V15 | 0.782 | 0.571 | 0.724 | 0.361 | 19.551 |
| S2-L1V15 + Dense | 0.941 | 0.587 | 0.725 | 0.397 | 17.183 |

(a) l31v32+dense15-17.  (b) l31v32+dense14-18.  (c) l31v32+dense12-20.  (d) l4v24+dense10-22.

Figure 6: 128k Needle In A Haystack Evaluation

## 5.2 Long Context Continual Pre-training

In this section, we further examine how to adapt sparse attention to longer contexts. Following the settings in previous section, we extended the best 1.3B model, S2-L1V15 + Dense, to 32k context length. We also continual pre-train Llama 2 to 128k context. We respectively changed the RoPE base to 1,00,000 and 5,000,000 for the 1.3B model and 7B model. Both models are continual trained with 10B tokens following the recipe in Fu et al. (2024). We evaluate the extended models on the Needle In A Haystack (Kamradt, 2023).

As shown in Figure2b, the 1.3B model can achieves perfect recall on the 32k context window without any further changes. For the 7B models, however, we need to tune the parameters to get a perfect score. Restricted by timing and computation resources, we mainly alternate the dense layer numbers and insertion depth, as we believe they are the quickest way to minimize the gap between sparse and dense architectures. As shown in Figure6, for 128k context, with a vertical stride size of 32, the model can retrieve the full context with 8 dense layers but fails to do so with only 2 and 4 dense layers. For a model with vertical stride of 24, the model can get perfect score with 12 dense layers.

The results validate the long context capability of S2-Attention design. We believe the density could be further trimmed given more tuning, which we leave for future work.

## 5.3 Training Speed-up

### 5.3.1 Attention Operation Benchmark

**Benchmark Settings** We measure the attention runtime of S2-Attention and FlashAttention-2 on an A100 80GB GPU for different context length, number of head, and head dimension settings.

In Figure7 and Figure2a We benchmark the speed-up brought by S2-Attention in 1.3B, 7B, 70B model sizes across different sequence lengths to showcase the scalability of our system. For all the

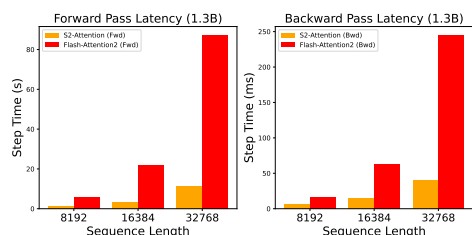
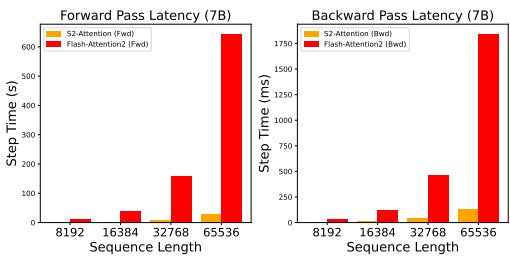

(a) 1.3B Attention Speed-up.        (b) 7B Attention Speed-up.

Figure 7: Attention Speed-up vs Sequence Length and Model Scale.

model sizes, S2-Attention can achieve multiple times of speed-up over FlashAttention-2. For 70B models with 64 heads, S2-Attention can give 25.3X end-to-end speed-up. For example, in 1.3B models with a vertical stride of 16, S2-Attention can achieve as much as 8.79X speed-up. As the max sequence length grows longer, the speed-up gradually approximates the theoretical FLOPs reduction benefits. The overall boost is hedged due to our less optimized backward kernel, which leaves room for further improvement.

### 5.3.2 END-TO-END TRAINING SPEED-UP

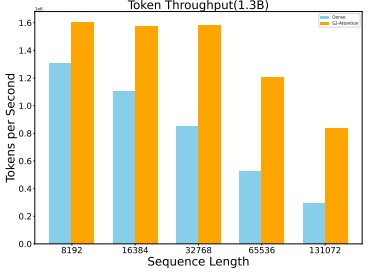
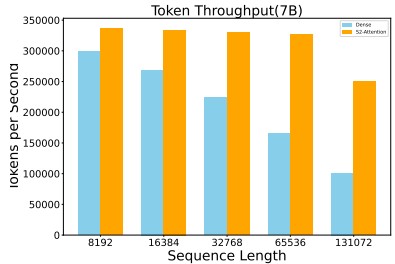

(a) 1.3B end-to-end training speed-up.        (b) 7B end-to-end training speed-up.

Figure 8: End-to-end training speed-up.

We evaluate the end to end training speed-up of the 1.3B and 7B models by measuring the token throughput of both models. All models are trained on 256 A100, with a batch size of 8M tokens, and activation checkpointing. For 1.3B models, S2-Attention can get 1.22X, 1.85X, 2.3X, and 2.83X token throughput on 8k to 128k context compared to FlashAttention-2. For 7B models, S2-Attention can get 1.12X, 1.24, 1.47X, and 2.47X token throughput improvement. Namely, we can half the time for training a 128k context length model.

### 5.4 INFERENCE SPEED-UP

In order to demonstrate the inference improvements of S2-Attention, we measure the end-to-end latency over different context length settings. To make comparison realistic, our experiments are done on vLLM (Kwon et al., 2023). We choose the FlashAttention-2 backend in vLLM as baseline for fair comparison, as the inference kernel of S2-Attention is also based on vLLM. Both methods are deployed on a single node with 8 A100 80GPU, with tensor parallel size equals 4. We set output length as 128 and vary input length between 16k to 256k. As shown in Figure 9, S2-Attention can achieves 1.1X, 1.2X,1.5X, 1.9X, 2.9X, 4.5X speed-up on 8k, 16k, 128K, 256K context.

### 5.5 ABLATION

In this section, we ablate a few heuristics for a better understanding of effective sparsity design. We do so by starting from the same S2-Attention with 8 local shards and a vertical stride of 16 shard. We then compare the validation perplexity and passkey result after they are trained with 300B tokens.

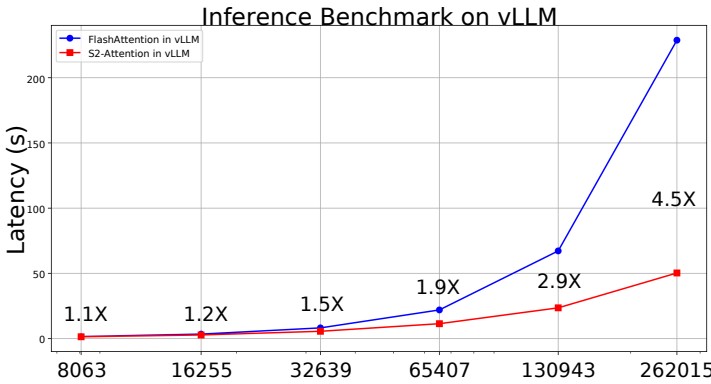

Figure 9: End-to-end latency speed-up for 7B size Llama architecture models.

### 5.5.1 DENSE LAYER

Table 2: Number of dense layers and insertion location Ablation

| Model | L8V16 | +Dense1,2 | +Dense23,24 | +Dense12,13 | +Dense12,18,24 | +Dense8,16,24 | +Dense18-24 |
|-------|-------|-----------|-------------|-------------|----------------|---------------|-------------|
| PPL | 11.72 | 11.49 | 11.51 | 11.49 | 11.49 | 11.49 | 11.82 |
| Passkey | 0.620 | 0.840 | 0.916 | 0.945 | 0.964 | 0.923 | 0.824 |

We first study the effect of dense layers as shown in Table 2. We can see adding dense layers can usually directly improve the validation perplexity and passkey results, except when we put all the last 6 layers dense. Different from previous studies Huang et al. (2022), where attention layer are more dense at the start and last layers, we find it's actually more beneficial to put dense layers around the middle during pre-training.

### 5.5.2 RETRIEVAL HEAD

Table 3: 7B pre-training results. S2 refers to S2-Attention. L refers to number of local blocks. V refers to the vertical stride size.

| Model | L8V16 | +Retrieval Head | +2*Retrieval Head |
|-------|-------|-----------------|-------------------|
| PPL | 11.72 | 11.61 | 11.61 |
| Passkey | 0.620 | 0.880 | 0.865 |

Wu et al. (2024) propose the idea of retrieval head, which are attention head scattering attention around the context. Retrieval head is found to be crucial to long context understanding tasks, and costs fewer compute and memory compared to dense layers. In Table3, we find adding 1 retrieval head can significantly improve the passkey performance and beneficial to perplexity. However, adding more of them doesn't further improve the model quality.

## 6 CONCLUSION

In this paper, we introduce S2-Attention, a framework that sparsely shards the context for different attention heads to divide-and-conquer. Experimental results show model trained with S2-Attention can achieve promising performance on long context tasks with reduced context for each head. We back S2-Attention with a highly optimized kernel library, we can get equivalent model quality while achieving speed-up over FlashAttention-2 linearly increasing over the number of attention heads. We open-sourced our kernel library and make it a plug-in-and-play alternative for FlashAttention-2 module in popular training frameworks like Megatron and Pytorch. We also integrated S2-Attention into vLLM backend for instant serving. Both the training and inference kernels allow users to freely customize their sparsity pattern, facilitating the whole community to study the topic in the future.

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

# 7 APPENDIX A: MODEL CONVERGENCE

To understand if trainings of different attention patterns converge similarly, we design experiments with dense attention, homogeneous and heterogeneous S2-Attention. Results are shown in Figure 10. We observe that the model with heterogeneous S2-Attention has almost identical loss curve as the dense attention one, showing same convergence. On the other hand, the model with homogeneous S2-Attention clearly shows a disadvantage in loss.

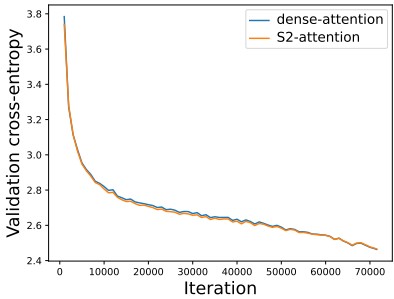 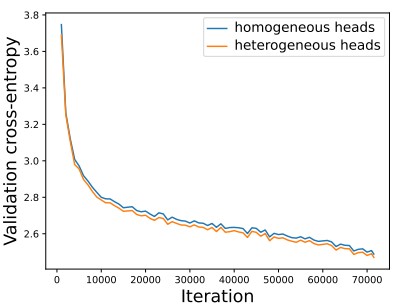

Figure 10: Convergence among dense attention, homogeneous and heterogeneous S2-Attention. **Left**: dense attention vs heterogeneousS2-Attention. **Right**: homogeneous vs heterogeneous with complete shards. All experiments are done with identical setting except the attention patters. For S2-Attention, local window = 8, vertical stride = 16, shard size = 64 tokens.

