# OpenReview forum: "S2-Attention: Hardware-Aware Context Sharding Among Attention Heads"
_ICLR.cc/2025/Conference — Submitted to ICLR 2025_

### Official Review · Reviewer_WoeU · 2024-10-28

**Soundness:** 2
**Presentation:** 2
**Contribution:** 3
**Rating:** 5
**Confidence:** 3

**Summary:**

This paper introduces Sparsely-Sharded (S2) Attention, a Triton-based library that optimizes sparse attention mechanisms for large language models (LLMs) by selectively attending to subsets of tokens. It proposes sharding context heterogeneously across attention heads to achieve higher parallelism and memory efficiency while maintaining strong downstream performance. Additionally, the paper demonstrates that hybrid architectures combining sparse and dense attention are particularly effective. Extensive experiments showcase the proposed attention mechanism’s significant wall-clock speedups and accuracy comparable to dense attention methods.

**Strengths:**

**Significant Performance Gains**: S2-Attention achieves impressive wall-clock speedups (8.79x, 15.87x, and 25.3x), outperforming FlashAttention-2 and maintaining strong downstream performance.

**Memory and Computation Efficiency**: The method offers higher parallelization and optimized memory I/O, crucial for large-scale models, especially when handling high token lengths.

**Real-World Applicability**: The library is designed to integrate with popular LLM frameworks (e.g., Megatron, vLLM) and provides user-friendly APIs, enhancing its usability for practitioners.

**Weaknesses:**

**Lack of Performance Comparison on Larger Models**: While the paper demonstrates hardware benefits on large-scale models, especially with high token lengths, it lacks the corresponding task performance comparisons.

**Paper Organization and Writing Issues**: There are several grammatical issues and unclear figure descriptions, which hinder the readability and understanding of the paper. For example:
1. The caption of Figure 5 only explains Figure 5(a) and lacks descriptions for Figure 5(b).
2. The legends in Figure 8 are too small to read. Additionally, there is an issue with the label on the y-axis of Figure 8(b).

**Questions:**

1. Although this paper highlights its advantages for large-scale models, especially with high token lengths, it only provides task performance results on Llama2-1.3B with a maximum sequence length of 8192, which somewhat limits the validation of its overall effectiveness.

2. Table 1 lists various S2-Attention variants, each with different task performance. Could the authors clarify which S2-Attention variant was used to measure the speedup and throughput in the subsequent experiments?

3. The paper reports both backward latency and end-to-end training speedup, validating its effectiveness for a single step. However, I would like to know if the sparse models proposed here require the same number of training steps to converge as dense models, so that the benefits extend to the total training time as well.

4. Since memory reduction is another key advantage of sparsity, it would be helpful to include comparisons in terms of GPU memory reduction for the proposed S2-Attention.

---

> ### Author Response · Authors · 2024-11-28
> **Response to Reviewer WoeU**
>
> Weakness 1:
> Besides the 1.3B model, we pre-trained two 7B models on 3T tokens, as well as four models for the continual pre-training setting.
> We first report the pre-training results here. We can see the S2-L8V16 and S2-L8V16-Inter both achieve strong performance.
>
> | Model           | MMLU  | HellaSwag | WinoGrande | PIQA  | BoolQ |
> |------------------|-------|-----------|------------|-------|-------|
> | S2-L8V16        | 56.3  | 75.7      | 78.6       | 80.7  | 81.7  |
> | S2-L8V16-Inter  | 57.6  | 78.7      | 70.3       | 70.6  | 83.4  |
>
>
> We also resent the downstream task performance of continual pre-training Llama 3.1 in addition to the 128k needle retrieval performance. We can observe all the downstream remains on par with the dense version.
>
>
> Weakness 2:
>
> We proof-read the paper and clear all the typos.
>
> Q1:
>
> A1: We eval 1.3B with context length extension to 32k, which has perfect needle score as shown in Figure 2(b). As we don’t do post training, it’s hard to do further on more complex tasks. For the 7B models, we trained on 128K context and got  perfect scores as shown in Figure 6.
>
> Q2:
>
> We thank the reviewer for pointing out the missing information that we lost at refactoring.  We’ll add them to the paper.  But here are  the details.
> For studying the speed-up of MergeQ and SplitD to attention-op ,  the experiments are conducted on a 1.3B model scale.  (batch size, heads, head dim)=(4, 16, 128). In sparse block size is 16 for merge-Q experiments and 64 otherwise. Vertical stride is 16 for merged-Q while varied otherwises. Local windows are all kept to 512 tokens.
> As for the end-to-end training and inference speed-up, for 1.3B models, the speed-up benchmark is done with L1V15 and 64 block size.
> For 7B models, the e2e speed-up benchmark is done with L4V24+dense10-22 with 64 block size .
>
> Q3:
>
> As we observe on-par converging speed for sparse models and dense models, the token throughput can actually be translated to end-to-end training speed-up. The sample training curve is shown in the Appendix A.
>
> Q4:
>
> Yes. At decoding, we roughly save (1-(1-local_window_size)/stride_size) of  memory.

---

> > ### Author Response · Authors · 2024-12-04
> > **Response to Reviewer WoeU**
> >
> > We thank the reviewer to honestly enumerate his/her concerns, which helps us fully explain our motivations and solutions. Research is meant to find real problems and create solutions to the problem so that the wider community can benefit from this new knowledge. In S2-Attention, we find an elephant in the room that is overlooked by our community, and try to point out a possible solution, which we carefully derisked on one of the largest LLM training and serving systems. We also open-sourced our kernel library as well as the cookbook for how to make sparsity work in training and inference. Some insights may be less straightforward for people with less system background and industry experience, but we tried our best to explain them.
> >
> > If our response helps clarify your concerns, we hope you can raise the score so that our knowledge can be better recognized.

---

### Official Review · Reviewer_rPC5 · 2024-11-02

**Soundness:** 2
**Presentation:** 2
**Contribution:** 2
**Rating:** 3
**Confidence:** 3

**Summary:**

This paper first introduces a Triton library that implements sparse attention customizable at both per-head and per-context-range levels, and then proposes Sparsely-Shardded (S2) Attention which shards the context heterogeneously across attention heads but collectively covers the full context. The evaluation show that the proposed S2-Attention can achieve 2.5X training speed-up and 4.5X inference speed-up for a 7B transformer architecture design.

**Strengths:**

+ this work presents a flexible kernel implementation that supports finer-grained sparse attention. Previous work FlashAttention-2 requires the sparsity granularity to be same as the block size, while this work introduces Merge-Q technique to effectively decouple the granularity of sparsity pattern and attention computation while achieving the expected speedup.
+ this work provides a detailed accuracy comparison to demonstrate the effectiveness of heterogeneous context sharing and union completeness.

**Weaknesses:**

- S2-Attention requires training models from scratch, raising concerns about its compatibility with pre-trained models. This limits its flexibility compared to other sparse attention methods  (e.g., QUEST, H2O) that support plug-and-play integration.
- the benefits of supporting finer-grained sparsity remain unclear; if existing block sparse attention methods suffice, the proposed library may be less practical.

**Questions:**

My questions are listed in the weakness section.

---

> ### Author Response · Authors · 2024-11-28
> **Response to ReviewerrPC5**
>
> Weakness 1:
>
> We thank the reviewer for this suggestion. One major point of this paper is stating that plug-in-and-play KV eviction methods do not work well with the underlying working mechanism of the hardware, primarily due to suboptimal GPU memory access and allocation. Therefore, they typically cannot bring wall-clock speed-up when compared to real serving systems like vLLM, SGLang, FlashInfer, and TensorRT, nor can they be easily implemented with these systems due to incompatibility with foundational techniques such as continuous batching and PagedAttention.
>
> More specifically, from a memory management perspective, (1) frequent manipulating small chunks of memory at arbitrary location, (2) different workload for each head/layer will cause significant memory fragmentation, instead of reducing the memory footprints. However, (1) and (2) are exactly what many of the KV eviction methods are doing. From a computational perspective, (2) will also hurt the benefit of parallelism and can hardly beat vanilla GEMM in reality.
>
> Because of these, the claimed flexibility of KV eviction methods, e.g., H2O and Quest, is only theoretical and can hardly be used in production with today’s accelerators. In fact, they are not adopted in any major LLM providers, nor the open-source community like vLLM.
>
> We propose S2-Attention aiming to bridge the gap between innovations in the research context and their real-world deployment. Meanwhile, S2-Attention is designed as a general attention alternative, instead of solely targeting for inference optimization. We actually experiment it for pre-training and context extension scenarios and observe on par results with dense attention and much faster training speed.
>
>
> Weakness 2:
>
> We thank the reviewer for pointing out this. We optimize the performance of finer-granularity because it’s an obvious feature that can help the community to explore sparsity more freely. Putting it straight, it’s similar to how improving magnifying glass to microscope would help science discovery.
>
> For example, the KV eviction methods usually need token-level sparsity. If now the system restricted users can only mask/keep every 64 tokens, which is the case to be compatible with FlashAttention, the exploration and study would be largely limited.

---

> ### Comment · Reviewer_rPC5 · 2024-11-29
> **Response to Authors' Rebuttal**
>
> Thank you for your explanation. However, your claim that prior sparse attention work, such as Quest, is limited to theoretical speedups and unsuitable for production is incorrect. FlashInfer [1] has already implemented block-wise sparse attention. In fact, prior works, including Quest and MInference, have demonstrated measured speedups using either the CUDA API provided by FlashInfer [2] or customized CUDA kernels. Ignoring these implementations misrepresents the state of sparse attention research. Therefore, I strongly agree with Reviewer 8jB7 that this work must include comparisons with other optimized sparse attention kernels tailored for prefilling (e.g., MInference) and decoding (e.g., Quest).
>
> Moreover, since S2-Attention requires training from scratch, the absence of performance benchmarks against dense models on diverse tasks and datasets, such as GSM8k and LongBench, is a critical omission that weakens the evaluation of the proposed method.
>
> [1] https://github.com/flashinfer-ai/flashinfer/issues/367
> [2] https://github.com/mit-han-lab/Quest

---

> ### Author Response · Authors · 2024-12-03
>
> Thanks for bringing up Quest. I think there is some misunderstanding on the paper.
>
> 1. This paper is not just about inference or KV-cache saving and inference speed. It is also about new pretraining architecture. Our method and kernel work for both training and inference, while the FlashInfer can only do inference.  FlashInfer benefits inference, while ours benefits both at training and inference. Therefore, the scope of our work is broader.  We enjoy the optimization done by FlashInfer, but in my understanding, it currently doesn’t support head-heterogenous sparsity patterns, nor usage in training.
>
> 2. From an industry perspective, training from scratch/requiring continual training is not a disadvantage. On one hand, plug-in-and-play is nice, but it probably not the top priority in real-world LLM deployment. What the real-world deployment will avoid the most is performance degradation and unpredicted model behavior, which KV eviction or Quest like methods usually suffer from due to the inconsistence between training and inference. The cost of regression in production is a lot more severe than the extra training before deployment.
>
> 3. For comparison with Quest:
>
>     a.) Quest does not save memory. Its implementation still needs to store the entire KV cache, plus the metadata for topk computation. It only saves memory bandwidth and computation (the so-called memory movement, i.e., loading from HBM to SRAM). Quest is actually motivated by the IO overhead and memory fragmentation issues in H2O, but trade-off the memory saving to avoid these overhead. However, S2-Attention’s design can achieve both targets. Like H2O, our method also saves memory for storing the KV-cache. Please note that for a single sequence of 128k input in a small 7B model, takes up 64GB of memory, meaning that you can only handle 1 user input at a time with a 80GB GPU (or 4 users input of length 32k), which is quite expensive. This is also part of the reason why Quest limits the batch size on each device to 1.
>
>     b.) In terms of computation, with a kv-cache budget of 2k in a 32k input, Quest achieves a speed up of 4.86x on attention, with 30% is for finding the top-k (without that, it is 7.03x ) as from Figure 9 of their paper. However, for S2-Attention, with a 2k budget out of 32k input (sparsity of 1/16), our speed up 2.3X over Quest. Most of the difference comes from Quest’s large overhead and the memory fragmentation. GPUs work better if loading a consecutive chunk, but top-k does not guarantee that even with paged KV. That’s what we mean hardware-aware.
>
>     c.) Another problem of Quest, is that the K in their “lossless” top-k budget in their benchmark changes through tasks. However, in a real application, we typically don’t know what task the user’s input is. S2-Attention don’t have this issue as inference and training are the same.
>
> Quest’s downstream benchmark is mostly on Longbench, but Longbench for a lightly pre-training model will not be great for benchmark (see point 3 below). So it is hard to compare the latency by controlling the accuracy (also due to point c above) or compare accuracy by controlling latency(our method could attend to almost 2x more tokens, with pre-trained backs the attention pattern, as well as memory saving) . However, from all the points above plus the results of downstream tasks in our paper, it is almost obvious that our method will likely balance latency and accuracy better, beside that fact that it also serves as a much faster pre-training method.
>
> 4. Regarding the GSM8k, LongBench, or even MMLU.  For pretrained models using open-source web data(e.g., fineweb-edu), it is challenging to achieve a sensible number for these tasks within hundreds of billions of tokens (already very expensive, 256 A100, a bit more than one day). We’d  probably need to train trillions of tokens to get sensible scores for a single model variation, which makes paper on pre-training prohibitive, let alone doing lots of ablations. Another way is to use non-open source synthetic data, e.g., Phi-3, which hurts reproducibility and is way too expensive for most researchers.  Indeed, cross-entropy or PPL is a very good metric and correlates well with downstream performance in realistic pre-training research, as long as the tokenizer and data are the same. That’s why many pre-training papers[1,2] only include loss and the tasks we presented on the paper.
>
> I hope the reviewer understands that this is not “inference/kv-cache” centric paper and the cost & limitation on publishing paper related to pre-training. The amount of exploration, insights, and invested resources are way beyond.
>
> [1] Yoco Neurips 2024: https://openreview.net/forum?id=25Ioxw576r&referrer=%5Bthe%20profile%20of%20Shaohan%20Huang%5D(%2Fprofile%3Fid%3D~Shaohan_Huang1)
> [2] Mamba: https://openreview.net/forum?id=AL1fq05o7H

---

> > ### Author Response · Authors · 2024-12-04
> > **Response to ReviewerrPC5**
> >
> > We thank the reviewer to honestly enumerate his/her concerns, which helps us fully explain our motivations and solutions. Research is meant to find real problems and create solutions to the problem so that the wider community can benefit from this new knowledge. In S2-Attention, we find an elephant in the room that is overlooked by our community, and try to point out a possible solution, which we carefully derisked on one of the largest LLM training and serving systems. We also open-sourced our kernel library as well as the cookbook for how to make sparsity work in training and inference. Some insights may be less straightforward for people with less system background and industry experience, but we tried our best to explain them.
> >
> > If our response helps clarify your concerns, we hope you can raise the score so that our knowledge can be better recognized.

---

### Official Review · Reviewer_yDhN · 2024-11-04

**Soundness:** 3
**Presentation:** 3
**Contribution:** 3
**Rating:** 6
**Confidence:** 3

**Summary:**

This paper introduces S2-Attention, a Triton-based library designed to optimize sparse attention in large language models (LLMs). Traditional sparse attention methods reduce FLOPs but fail to deliver real-world speedups due to memory access inefficiencies. S2-Attention addresses this with customized kernel optimizations, enabling flexible and efficient sparse attention patterns at per-head and per-context levels.

Key contributions include:
1. S2-Attention uses a novel method of context sharding across attention heads, maximizing parallelization and improving memory access efficiency.
2. By combining sparse and dense layers, S2-Attention achieves performance on par with dense attention models while accelerating training and inference.
3. The library achieves up to 25.3x speedup over FlashAttention-2 in 7B models and provides substantial efficiency gains across different model sizes and context lengths.

**Strengths:**

1. The paper presents a novel approach to improving the real-world efficiency of sparse attention mechanisms in LLMs through S2-Attention, a customizable, hardware-optimized library. Unlike prior sparse attention methods that often fail to deliver actual speedups, S2-Attention effectively addresses the GPU memory access bottleneck. Additionally, the hybrid architecture combining sparse and dense layers is an innovative solution to balance efficiency and model performance.
2. The paper demonstrates high-quality research with extensive benchmarking and thorough evaluations that underscore S2-Attention's performance benefits. The experiments cover a range of model sizes (e.g., 1.3B and 7B models) and contexts, showing both training and inference gains across different configurations. The authors also offer a clear implementation path in Triton, indicating that the work is robustly engineered for real-world applications.
3. By achieving up to 25.3x speedup over FlashAttention-2 and providing easy-to-use APIs for integration into frameworks like Megatron and vLLM, S2-Attention has the potential to become a standard tool for optimizing sparse attention in LLMs.

**Weaknesses:**

The paper is innovative in its approach and thorough experimentation. However, there are several critical questions that I raised in the "Question" section, which I believe are essential for the clarity and robustness of the findings. I hope the authors can provide insights on these points, and I look forward to further discussion.

**Questions:**

1. I am also a researcher specializing in hardware optimizations for Transformers, and I recognize the importance of sparse attention. Due to the self-attention mechanism in Transformers, there is significant redundancy since token-to-token similarities vary in each head’s activation, and sparse attention can help reduce this. However, because operations across heads are independent in Transformers, applying different attention sparsity patterns to each head does not traditionally contribute much to overall model acceleration. Each head has a different computational overhead, leading to an imbalance. Ultimately, the runtime is dictated by the head with the highest computational complexity. In Figure 1, the 'S2-Attention' subfigure seems to illustrate this issue, but it doesn’t appear that the authors discuss this challenge in the library design. Could the authors clarify if this imbalance was considered and, if so, how it was addressed?
2. In Section 3.2, the authors mention that "retaining or masking every 64 tokens" allows the dense portion of sparse attention to fill each block. However, in Section 3.1, they state that "each warp contains 32 threads." The values of 32 and 64 seem contradictory. Is there a relationship between the 64-token blocks and the 32-thread warps, or are these independent design choices?
3. In Figure 2(b), the authors seem to indicate that with a 32K context length, the “depth perfect” remains at 100% regardless of the token limitation. How is “depth perfect” defined in this context? It doesn’t appear to be referenced or defined elsewhere in the paper. Could the authors clarify? The same question for Figure 6(c)(d).
4. In Figure 3, what is the control logic for data movement in MergeQ? Does it follow a fixed pattern? What kind of memory control logic is used in the SRAM for this process? Are the authors implementing in-memory computation within the SRAM, and is the softmax operation also executed within SRAM? Could you please provide a more detailed explanation of the MergeQ process, perhaps with a step-by-step breakdown or a flowchart illustrating the control logic and data movement?
5. Lastly, could the authors clarify how many hardware measurements were conducted to obtain the reported results? Given the inherent variability in hardware performance, multiple measurements are typically necessary to ensure stability and reliability. It would be helpful to understand the stability of the hardware results presented. Could you please provide details on the number of runs performed for each experiment, any measures taken to account for variability (e.g., averaging results, reporting standard deviations), and how you ensured the stability and reproducibility of your hardware measurements?

---

> ### Author Response · Authors · 2024-11-28
> **Response to Reviewer yDhN**
>
> Q1:
>
> Thanks for the great question. It is true that the FLOPS among heads are imbalanced. But it is not true that the computation is decided by the most intensive head. The parallelization is not just on head level, but also on token chunk level, i.e., the computation on one-head is not sequential. The q will be chunked and parallel (indeed, even kv can also be chunked and parallel, but mostly we typically don’t find it necessary in training but only at inference).  So you see that for a head, the most intensive one is the last chunk (attend to the most kv).  Therefore, over all heads, the most intensive threadblock is the one handling the last chunk of the least-sparse head. But chunks close-by have similar computations as well. Remember that GPU’s assignment of threadblock to SMs is random and based on availability, so as long as the number of thread blocks is high (significantly higher than SMs), it will even out. This is the case in length like 4k (32 heads x 4k/64 block size=2k thread block >> 106 SMs in A100).
>
> In addition, the patterns (local-stride) used in the paper (Figure 1), all mostly balanced (but not required even from computation perspective as mentioned above).  We did try mixing sparse heads with dense heads as well, and we don’t observe computation downside due to head imbalance. We didn’t include it in the paper due to length.
>
> Q2:
>
> On an NVDA GPU, a warp consists of 32 threads (fixed) , there are threads that executes together.  Warp and thread blocks are separate concepts. In our code, a block of 64 tokens is computed in a threadblock, which can contain 1, 2 or more warps (it is a hyperparameter to tune). The mapping of data and warps is complex. A warp (32 threads) can handle 16 tokens, 32 tokens, 64 tokens or even more. There is no such a restriction of 32 threads handling 32 tokens, but rather based efficiency consideration.
>
>
> Q3:
>
> For the 32k length, we continual pretrain the 1.3B model whose max sequence length was 8192 to get context extension. We follow this training protocol as previous studies suggest. “depth perfect” means the extended model can perfectly retrieve the information, namely “needle”, wherever the needle is put in a 32k context.
>
>
> Q4:
>
> Yes, we follow a fixed, rule-based pattern for MergeQ. As long as the vertical stride pattern is used, we can easily decide the SRAM access logic. We merged q-blocks when sparse block size are smaller (e.g., 16 tokens)  than efficient kernel block size (64).   Softmax is fused in an online way similar to FlashAttention.
>
> Q5:
>
> For the forward/backward figures, it is 100 times for benchmark, with 25 warm-up in advance using “triton.testing.do_bench”.
> For inference in vLLM, we use the default, 30 times with 10 times of warmup in advance.  The warm-up here is mostly for optimization like tuning(Triton) or cudagraph(vLLM) building.
>
> In all those tests, we found that p50, p95 and average are almost identical (<1%). Therefore, we don’t bother to report the standard deviation.  For GPU (A100, H100, not those for displaying), as long as it is solely occupied by the benchmarking process, it seems to be very stable across runs, unlike CPU which is never completely free. That’s why in FlashAttention 1 and 2 papers, no standard deviation was mentioned either.

---

> > ### Author Response · Authors · 2024-12-04
> > **Response to Reviewer yDhN**
> >
> > We thank the reviewer to honestly enumerate his/her concerns, which helps us fully explain our motivations and solutions. Research is meant to find real problems and create solutions to the problem so that the wider community can benefit from this new knowledge.
> >
> > If our response helps clarify your concerns, we hope you can raise the score so that our knowledge can be better recognized.

---

### Official Review · Reviewer_8jB7 · 2024-11-04

**Soundness:** 2
**Presentation:** 2
**Contribution:** 2
**Rating:** 3
**Confidence:** 4

**Summary:**

This paper presents a Triton-based GPU kernel library designed to enhance the efficiency of sparse attention training and inference. By merging query blocks and splitting along head dimensions, the library improves GPU warp utilization, particularly for fine-grained sparse attention patterns. With this library, the authors propose a KV-cache-efficient heterogeneous sparse attention method, showing the performance and efficiency benefits of the library.

**Strengths:**

1. Useful libarary. The paper implements a practical sparse attention GPU kernel library that supports both training and inference. The flexibility to support fine-grained sparse patterns can benefit future research towards more effective and efficient sparse pattern design.
2. High efficiency. With the optimized sparse attention kernel, the paper shows speedups of up to 25.3 and 4.5 times for training and inference over the dense FlashAttention baseline.

**Weaknesses:**

1. The main concern of the paper lies in the proposed sparse attention pattern design. The proposed KV-Cache design principle seems overly conclusive and conflicts with existing works.

    a. The principle itself is not novel; similar sparse pattern designs for KV-Cache optimization have been explored extensively in prior studies, such as [1, 2]. Furthermore, recent work on retrieval-based KV-Cache reduction [3] demonstrates high performance despite contradicting this principle. It would be beneficial for the authors to revise their claims to improve rigor and acknowledge alternative approaches.

    b. The deduction of the claim "the sparse patterns should be based on absolute positions rather than relative ones" is confusing. It is unclear why the "vertical line" sparse pattern leads to absolute positions, and why windowed attention is the only exception.

2. The advantages of the proposed heterogeneous context sharding pattern over existing designs are not clearly shown. Additionally, it is unclear how this pattern adapts to varying input lengths, as the ranges of context shards appear pre-defined and fixed.

3. The performance comparison with other sparse attention kernels is not shown. Beyond dense attention methods, it would be valuable to assess the proposed kernel’s performance against other sparse attention methods with GPU kernels, such as those optimize prefill [4] and decode [5].

4. Minor writing issues:

    a. Gramma and typo: Spelling mistake in section 3.1: imrpvoes -> improves; in section 5.1: th -> the

    b. Clarity: In the abstract, it should be specified that the “8-25x speedup” refers to training time.

[1] Xiao, Guangxuan et al. “Efficient Streaming Language Models with Attention Sinks.”, ICLR'24
[2] Zhang, Zhenyu (Allen) et al. “H2O: Heavy-Hitter Oracle for Efficient Generative Inference of Large Language Models.”, NeurIPS'23
[3] Tang, Jiaming et al. “Quest: Query-Aware Sparsity for Efficient Long-Context LLM Inference.”, ICML'24
[4] Jiang, Huiqiang et al. “MInference 1.0: Accelerating Pre-filling for Long-Context LLMs via Dynamic Sparse Attention.”, arXiv'24
[5] Fu, Tianyu et al. “MoA: Mixture of Sparse Attention for Automatic Large Language Model Compression.”, arXiv'24

**Questions:**

1. How does the efficiency of the proposed sparse attention kernel compare to that of other sparse attention methods?
2. In what ways does the proposed sparse attention pattern adapt to varying input lengths, and what is its performance across different lengths?
3. Given that the kernel dedicatedly optimized for fine-grained sparse patterns, how does the fine-grained sparse pattern impact model accuracy? What are the performance-efficiency trade-offs at different granularities for the proposed kernel compared to others?

---

> ### Author Response · Authors · 2024-11-12
> **Response to Reviewer 8jB7 about Weakness 1**
>
> Overall response to Weakness 1: Despite their conceptual elegance, previous KV cache reduction techniques often fail to translate the efficiency gains on paper to real-world applications. This is because, similar to FlashAttention vs Approximate Attention, the main bottleneck for sparse inference is memory management overhead and lack of computation kernel. S2-Attention aims to bridge the gap between innovations in the research context and their real-world deployment.
>  Weakness 1.a: We thank the reviewer for correctly pointing out that our work draws a conclusion that differs from some previous works [1]-[5]. A key insight of our work is that many  previous KV cache reduction methods, including those cited by the review, are often incompatible with the underlying working mechanism of the hardware used for serving these models. As a result, their promising savings on paper rarely translate speedup in practice and are not adopted in industry providers like OpenAI/Microsoft and Anthropic. This is the precise reason that, to the best of our knowledge, none of them are implemented in, or compared to, actual inference optimization systems widely used in practice such as, vLLM, SGLang, FlashInfer. Therefore, we consider our findings on the limitations of these prior works to be novel, as is the S2-Attention architecture, which is designed based on the principles learned from these limitations. If the reviewer can provide evidence contradicting our claims, we are happy to revise them. Otherwise, we respectfully disagree with the review’s concerns about the rigor of our claims and kindly ask the reviewer to reconsider their evaluation.
>
> About weakness 1.b: Those are great questions. Maybe we didn’t convey the message very well, due to length constraints.
>
> The key principle here is that efficient sparsity should avoid memory fragmentation, which [1]-[5] and most of the KV eviction works overlook. For example, it's not feasible to release/allocate small chunks frequently, e.g., 2 bytes, of memory at arbitrary, non-deterministic locations decided on GPU, as many of these works do. It also won't save memory, but only cause more memory fragmentation, e.g., by having different sparse attentions across instances in the same batch.  "Absolute positions rather than relative ones" ensures that the deterministic pattern among instances and avoids memory fragmentation. "Vertical line" means a KV block should be kept for a few decoding steps for more efficient memory management, compared to non-consecutive usages (fragmentation and memory waste) or immediate/frequent release/allocation (hurt performance).
>
> Take dilated attention as an illustrative example,  as shown in Figure 5a. Every token attends to 1 token every 2 tokens depending on the relative distance. Now token n attends to tokens n-2, n-4, …. So we need to cache (n-2, n-4, …) for it.  Let’s move on to token n+1, the next token. It requires the KV at tokens n-1, n-3,... Therefore, it requires caching all of them for decoding starting from token n, which ends up keeping all post tokens in memory despite not being used in many steps. This is  typical sparse attention that causes internal fragmentation.
> Moreover, runtime eviction like [1]-[5]introduces even more fragmentation in addition to what’s mentioned above. When the KV is different for each head or each layer, it will cause similar fragmentation to that caused by various sequence lengths in a batch as pointed in paged attention. Moreover, computation-wise, the workstream could be highly imbalanced and unpredictable, which hurt the benefit of foundational tensor/pipeline/sequence parallelism.
>
> This observation inspires us to study a KV-cache efficient pattern. For sliding-window-like attention, KV-cache is efficient despite that they are relative to token position. But for tokens outside of the sliding window, i.e., the “global” tokens attended by many tokens far away, attention patterns based on relative positions will cause fragmentation, as in the example above.
>
> That’s how we come to the definition of efficient kv-cache pattern,
> If (k_i , v_i) is attended by q_{j+l},  (k_i , v_i) must also be attended by q_j.  j >=i, l >0 .
> If you pause and think about it from the argument above, you can see that otherwise, we’ll have (k_i, v_i) at hand (as we need it for token j+l)  when we are decoding token j, yet we don’t use it, which isn’t “the best” for decoding token j as we have access to more information.
>
> This definition very naturally shows that token i above, has to be attended by a continuous segment of tokens after i before it is dropped completely. Obviously this is based on “absolute position” (position i) instead of relative position.  Hence the example patterns shown on Figure 5(b), where you see the “vertical lines”
>
> I hope this explanation of “kv-cache efficient pattern” makes things clearer. Due to the length of the paper, we didn’t include all these detailed arguments/steps.

---

> > ### Author Response · Authors · 2024-11-28
> > **Response to Reviewer 8jB7 about Weakness 2, 3, 4 and all the Questions 1, 2, 3**
> >
> > Weakness 2:
> >
> > We actually evaluate S2-Attention varying input lengths. Quality-wise, WinoGrande, PIQA, race are shorter ones ranging from a few dozens to a few hundreds of tokens as shown in Table 1. Passkey and Needle are measured from hundreds to 32k/128k tokens as shown in Figure 2 and Figure 6. They demonstrate the effectiveness of heterogeneous sharding. For speed, we benchmarked across context lengths up to 256k as shown in Figure 7, 8, 9. With these comprehensive experiments, we believe that we have provided extensive evidence showing that S2-Attention successfully adapts to varying input lengths. As for the advantages of our proposal over existing designs, we have already compared the strongest baselines that can achieve much better efficiency than KV eviction methods. Should the reviewer have more evaluations in mind, we are open to incorporating them if time and resource permit.
> >
> > Weakness 3:
> >
> > We thank the reviewer for suggesting these baselines. In fact, we already compare to baselines that are stronger than the one suggested by the reviewer. As mentioned in W1, [4][5] are even slower than the FlashAttention backend of vLLM, which we compare to in Figure 7 and  Figure 9 showing that our method is substantially faster. In addition, this paper has sparsity built into training instead of the ad-hoc solution in [4][5], which makes training faster and makes inference exact instead of post training ad-hoc/approximation.
> >
> > Weakness 4:
> >
> > We thank the reviewer for pointing out the typos and providing writing suggestions, which we have revised.
> >
> >
> > Question 1: How does the efficiency of the proposed sparse attention kernel compare to that of other sparse attention methods?
> >
> > Answer 1: We are much more efficient than (1) KV eviction methods, (2) previous sparse attention training methods, e.g. sparse transformer. For (1), all the mentioned methods are much slower than vLLM both in terms in latency and throughput. For (2), they're slower than FlashAttention, let alone FlashAttention-2. We demonstrate S2-Attention is multiple times faster than default dense attention in vLLM and FlashAttention-2 in the Figure 7, 8, 9, which makes our method much faster than the mentioned baselines. Please see our response to Weakness 1 and 3 for why S2-Attention is more efficient and why existing sparse attention methods are not. Moreover, most of the sparse methods mentioned are not compatible with actual training and serving systems used by researchers and companies, e.g., Megatron/Torch/vLLM, while S2-Attention is fully compatible to them and put into training and serving flagship LLMs in production setting for a while.
> >
> > Question 2: In what ways does the proposed sparse attention pattern adapt to varying input lengths, and what is its performance across different lengths?
> >
> > Answer 2: We benchmarked both the quality and performance over different context lengths, from 50-128k. Please see our response to Weakness 2 and  Sections 5.1, 5.2, 5.3.
> >
> > Question 3:
> >
> > Answer 3: Thanks for the great suggestion on exploring the trade-offs by varying block size. Our Triton implementation does allow changing the block size optimization to improve the generality and customization of the framework: it can support as small as a 16 block size We have not explored it since the goal of this work is not to seek a SOTA architecture due to time and resources limitations; rather, we aim to provide the community a flexible framework, e.g., supporting more customizable sparse masks. You can think of it as upgrading the magnifying glass to microscope for the biomedical researchers, which is intuitionally beneficial for the community. For S2-Attention, you can imagine if yourself would prefer a 64 mask-or-keep mask, or a smaller granularity one, e.g., 16. For the block size, S2-Attention provides16,32,64,128,256 and more.

---

> > ### Comment · Reviewer_8jB7 · 2024-11-29
> >
> > Regarding Weakness 1a, I suggest the authors use quantitative wall-clock speedup experiments or numeric results from previous papers to substantiate claims about the efficiency of previous works, rather than relying on descriptive conclusions. For instance, reference [3] reported a 2.23x self-attention speedup over FlashInfer, an “actual inference optimization system,” while [4] highlights the use of efficient sparse kernels. Including such numerical results would strengthen the arguments.

---

> ### Author Response · Authors · 2024-12-04
> **Response to Reviwer 8jB7**
>
> For [4], MInference, in many cases, is significantly slower than flash_attn backend of vLLM. The advantage is shown when the prompt is sufficiently long to cover the overhead, e.g., >100k.
> For numerical comparison, at 262015 prompt length, we're 2.28X faster than MInference. At 128k prompt length, we're 1.91X faster than MInference.
>
> For [5], Quest, with a kv-cache budget of 2k in a 32k input, Quest achieves a speed up of 4.86x speed-up on attention, with 30% is for finding the top-k (without that, it is 7.03x ). However, for S2-Attention, with a 2k budget out of 32k input (sparsity of 1/16), our speed up 2.3X over Quest. Most of the difference comes from Quest’s large overhead and the memory fragmentation. GPUs work better if loading a consecutive chunk, but top-k does not guarantee that even with paged KV. That’s what we mean hardware-aware. Meanwhile, Quest's bonus become significantly thinner when moving to A100/H100, we suspect the original speed-up partially came from the fact that FlashAttention/FlashInfer algorithms become weaker on commercial GPUs like RTX4090, which Quest benchmarked on. Also, [5] actually supports some of our main argument for those sparse KV, as [5] is motivated by the IO and fragmentation issues in H2O, but [5] trade-off memory saving for this purpose. S2-Attention's design can achieve both targets without trading-off memory saving for reducing IO and fragmentation overhead.
>
> Both [4] and [5] have issues for batch size > 1 on single device, and for correctly handling tensor parallel and pipeline parallel, which further weaken the practical of those systems. It also makes our comparison infeasible, as we can easily use TP=2 for S2-Attention to larger the throughput advantages over the [4][5]. As discussed in previous comments, dynamic KV eviction (though Quest doesn't evict), will have too many impractical overheads to be implemented in real serving systems.
>
> Another point is, both [4][5] needs to specify different hyperparameters for the framework to work and get the claimed speed-up. MInference get the minimum FLOPs needed for different tasks to maintain model quality. Similarly, Quest needs different K for their topK on different tasks. However, for realistic deployment where user query is unpredictable, it's not feasible to do so.
>
> We also need to stress that S2-Attention is not a "KV eviction/inference optimization only" paper. S2-Attention and the open-sourced kernel, are designed for both training and inference. We bridge the long-lasting absence of an optimized sparse self-attention framework, which can facilitate model architectures greatly. However, [4][5] and most of the inference optimization framework cannot do this. On the other hand, "plug-in-and-play" is not the top priority of actual LLM deployment and efficiency optimization in industry. We ourselves deployed and optimized several most widely used LLM endpoints in the world. The lessons we learned are (1) model quality regression, e.g., unpredicted behavior, is the worst thing we want to avoid, while any plug-in-and-play sparse module can cause degradation and change model behavior, (2) to the best of our knowledge, most of our counterparts and ourselves have the time and computation (continual)train a more efficient model variants, and are willing to do so as long as it's beneficial to long-term production benefits.
>
> [a] MInfernce degradation and imcompatibilities: https://github.com/vllm-project/vllm/issues/5751#issuecomment-2225698913
> [b] MInference TP failure:  https://github.com/microsoft/MInference/issues/63
> [c] MInference slower than vLLM: https://github.com/microsoft/MInference/issues/18
> [d] MInference performance degradation: https://github.com/microsoft/MInference/issues/83, https://github.com/microsoft/MInference/issues/43
> [e] QUEST TP: https://github.com/mit-han-lab/Quest/issues/8

---

> > ### Author Response · Authors · 2024-12-04
> > **Response to Reviwer 8jB7**
> >
> > We thank the reviewer to honestly enumerate his/her concerns, which helps us fully explain our motivations and solutions. Research is meant to find real problems and create solutions to the problem so that the wider community can benefit from this new knowledge.
> > In S2-Attention, we find an elephant in the room that is overlooked by our community, and try to point out a possible solution, which we carefully derisked on one of the largest LLM training and serving systems.
> > We also open-sourced our kernel library as well as the cookbook for how to make sparsity work in training and inference.
> > Some insights may be less straightforward for people with less system background and industry experience, but we tried our best to explain them.
> >
> > If our response helps clarify your concerns, we hope you can raise the score so that our knowledge can be better recognized.

---

### Meta-Review · Area_Chair_N3Nu · 2024-12-21

**Metareview:**

**Summary:** The paper introduces S2-Attention, a Triton-based library for hardware-aware sparse attention optimization, featuring context sharding across attention heads to enhance memory efficiency and parallelization. Using this library, the work highlights the effectiveness of a hybrid architecture that combines sparse and dense attention. Evaluations of S2-Attention with model sizes ranging from 1.3B to 7B demonstrate significant speedups compared to FlashAttention-2, while maintaining competitive performance on downstream tasks.

**Strength:**

1. The paper proposes a novel hardware-aware context sharding technique that effectively balances computation across attention heads, addressing the gap between theoretical efficiency gains of sparse attention methods and practical wall-clock speedups

2. It demonstrates notable wall-clock speedups compared to strong baselines, with practical applicability to long-context scenarios.

3. This work provides an open-source kernel library with user-friendly APIs, facilitating broader adoption in LLM training and inference pipelines.

**Weakness:**

1. Technical contributions and rationale: The proposed sparse attention pattern design lacks sufficient justification. As noted by Reviewer 8jB7, the principle behind the design is not novel. A more thorough analysis is needed to explain why the proposed design outperforms other designs within the existing design space.

2. Insufficient baseline comparisons: The paper does not provide adequate comparisons to other state-of-the-art sparse attention methods, such as Quest, MInference, and FlashInfer. These comparisons are essential to validate the claimed advantages of S2-Attention.

3. Training cost and applicability: S2-Attention requires training models from scratch, which significantly reduces its flexibility and limits its applicability compared to plug-and-play sparse attention methods. The paper does not sufficiently justify the additional training costs associated with this approach.

4. Clarity and writing issues: Grammatical errors, unclear figure descriptions, and inconsistent terminology detract from the paper's overall readability. Key aspects of the proposed method, such as the definition of "depth perfect" and the specifics of the MergeQ process, are not well explained, making the method less accessible to readers.

**Reasons for the decision:**

While S2-Attention provides a practical library for sparse attention, the lack of thorough analysis of the proposed attention pattern, missing baseline comparisons, and insufficient justification for the additional training costs diminish the paper's overall impact. Consequently, I recommend rejection.

**Additional Comments On Reviewer Discussion:**

During the rebuttal period, the common concerns raised by the reviewers and the corresponding author responses are provided as follows:

**1. Sparse attention pattern design (Reviewers 8jB7, rPC5):**

Reviewer Concerns: The reviewers criticized the novelty of the proposed sparse attention pattern, noting that similar principles have been explored in prior works. They requested more detailed analysis to justify the claimed superiority of the design over other approaches.

Author Response: The authors argued that their hardware-aware design offers practical benefits over existing methods by reducing memory fragmentation and improving parallelization. They provided additional explanations but did not include significant new comparative experiments to substantiate these claims.

**2. Baseline comparisons (Reviewers 8jB7, rPC5):**

Reviewer Concerns: Reviewers requested comparisons with state-of-the-art sparse attention methods, such as Quest, MInference, and FlashInfer, to validate the claimed advantages of S2-Attention.

Author Response: The authors provided additional insights into why some baselines were excluded, citing issues like compatibility and resource constraints. However, they did not conduct new experiments to address this gap.

**3. Training cost and applicability (Reviewers rPC5):**

Reviewer Concerns: Reviewers highlighted the need for models to be trained from scratch as a limitation, especially when compared to plug-and-play sparse attention methods. They questioned the practicality of the additional training cost.

Author Response: The authors argued that training from scratch is acceptable for production-scale deployments and emphasized the potential long-term benefits of S2-Attention’s pretraining-compatible design. They also clarified that their focus is broader than just inference optimization.

**4. Clarity and writing issues (Reviewers 8jB7, WoeU):**

Reviewer Concerns: Reviewers noted unclear figure descriptions, inconsistent terminology, and grammatical errors that hindered understanding. Specific concerns included the definition of “depth perfect” and the details of the MergeQ process.

Author Response: The authors revised the manuscript to improve clarity, fixed grammatical issues, and provided additional explanations for “depth perfect” and MergeQ.

While the authors made efforts to clarify their contributions and address some technical concerns, critical issues regarding technical novelty, baseline comparisons, and training cost justification remain unresolved. Concurring with the majority of reviewers who lean towards rejection, I also recommend rejecting this paper.

---

### Decision · Program_Chairs · 2025-01-22

Reject